# Maximizing Equity in Acute Coronary Syndrome Screening across Sociodemographic Characteristics of Patients

**DOI:** 10.3390/diagnostics13122053

**Published:** 2023-06-14

**Authors:** Gabrielle Bunney, Sean M. Bloos, Anna Graber-Naidich, Melissa A. Pasao, Rana Kabeer, David Kim, Kate Miller, Maame Yaa A. B. Yiadom

**Affiliations:** 1Department of Emergency Medicine, Stanford University, Palo Alto, CA 94304, USA; 2Tulane University School of Medicine, New Orleans, LA 70112, USA; 3Quantitative Sciences Unit, Stanford University, Palo Alto, CA 94304, USA

**Keywords:** acute coronary syndrome, ACS, screening, diagnosis, emergency, risk, prediction, equity, sensitivity, specificity, electrocardiogram, ECG, EKG, augment, predictive model

## Abstract

We compared four methods to screen emergency department (ED) patients for an early electrocardiogram (ECG) to diagnose ST-elevation myocardial infarction (STEMI) in a 5-year retrospective cohort through observed practice, objective application of screening protocol criteria, a predictive model, and a model augmenting human practice. We measured screening performance by sensitivity, missed acute coronary syndrome (ACS) and STEMI, and the number of ECGs required. Our cohort of 279,132 ED visits included 1397 patients who had a diagnosis of ACS. We found that screening by observed practice augmented with the model delivered the highest sensitivity for detecting ACS (92.9%, 95%CI: 91.4–94.2%) and showed little variation across sex, race, ethnicity, language, and age, demonstrating equity. Although it missed a few cases of ACS (7.6%) and STEMI (4.4%), it did require ECGs on an additional 11.1% of patients compared to current practice. Screening by protocol performed the worst, underdiagnosing young, Black, Native American, Alaskan or Hawaiian/Pacific Islander, and Hispanic patients. Thus, adding a predictive model to augment human practice improved the detection of ACS and STEMI and did so most equitably across the groups. Hence, combining human and model screening––rather than relying on either alone––may maximize ACS screening performance and equity.

## 1. Introduction

Using predictive models in clinical care has high potential to improve care quality and equity, especially as the numbers of evidence-based standards for diagnosis and treatment increase. Such improvements can only be achieved, however, when existing biases in practice and clinical data are understood and accounted for in clinical implementation [1]. We explore this dynamic within emergency care, specifically screening for acute coronary syndrome (ACS) to capture patients with ST-elevation myocardial infarction (STEMI).

Emergency departments (EDs) have screening protocols to identify patients who present on arrival with symptoms concerning ACS [2]. As per the guidelines adopted internationally, these patients should receive an electrocardiogram (ECG) within 10 min of arrival at the ED to identify the severe sub-diagnosis called STEMI [3,4]. An early ECG, performed within 10 min, allows clinicians to diagnose and treat STEMI quickly [5]. For every minute that STEMI diagnosis is delayed, the interventions are less effective, and the risks of heart failure and mortality are increased [4,6,7]. In current practice, ACS screening is often performed manually by non-clinical staff. Once they determine an ECG should be performed, it is immediately reviewed by a physician to determine the next step in the patient’s care pathway.

In order to ensure usability, current screening uses simple criteria such as age and chief complaint to encourage consistent application [6]. This approach has been noted to underdiagnose minority groups and women [8,9,10,11]. In particular, a prior investigation showed that Black patients have a 51% greater chance of delayed ACS diagnosis than White patients, and female patients have a 36% greater chance of delay than male patients [4]. Furthermore, oversimplified screening is unable to capture variations in risk that are needed for a more precise prediction of ACS, leading to disparities in timely interventions, including thrombolysis or percutaneous coronary intervention (PCI) for those with STEMI. This compromises a critical step in the STEMI chain of survival [12].

Prior work has shown that predictive modeling can augment current screening practices to shorten the time to diagnosis for more patients [13]. This was evidenced by higher sensitivity for identifying ACS and an increased proportion of those subsequently diagnosed with STEMI due to earlier screening without increasing the number of ECGs performed [7]. It is not clear, however, whether a model using the same data as manual practice will improve screening quality for all patients or exacerbate existing disparities for women and minority groups [14]. In addition, the literature suggests the best use of predictive models in screening and diagnostic care is to augment human performance, but this has not been tested for this clinical screening challenge [15,16,17].

To explore this, we evaluated four ACS screening approaches and their performance across sex, age, race, ethnicity, and language groups. First, we measured the proportion of patients in a large, urban ED who received ECGs within 10 min of arrival (observed human practice). Next, we simulated three alternative screening approaches in the same population. We asked what would have happened if the clinical protocols were applied perfectly, if a statistical predictive model screened for ACS, and if the observed human practice was augmented with the predictive model.

For ACS screening and STEMI diagnosis, missing any case is unacceptable [18,19,20]. Identifying those with ACS risk is the early clinical goal [4,6,20], so we took the sensitivity of screening approaches as our primary measurement. Our objective was to quantify differences in these screening approaches by demographic subgroup, with the ultimate goal of informing the discussion on how using predictive algorithms in clinical practice can alleviate bias.

## 2. Methods and Materials

### 2.1. Study Design

This was a simulation of comparative ACS screening approaches vs. manual screening that occurred within actual care. For this, we used a 5-year retrospective electronic health record cohort including emergency department (ED) visits at Stanford University from 1 January 2015 to 31 December 2020 at one large urban hospital. We obtained an ethics review via the Stanford Institutional Review Board before initiating the study and collecting data (IRB number: 56066).

### 2.2. Screening Approaches

ACS screening is designed to identify patients for an “early ECG,” which is an ECG within 10 min of arrival to the ED. A “positive screen” is a patient who is identified as requiring an early ECG. We restricted our screening approaches to using only the data elements that (1) are typically available during ED registration and (2) are part of this institution’s current screening guidelines: age, chest pain, and other ACS-associated symptoms [15]. We describe the method for applying the screening criteria below.

#### 2.2.1. Screen 1: Observed Human Practice

Registration clerks use arrival intake data such as age and chief complaint to determine whether a patient should receive an early ECG according to the ED’s screening protocol. To measure how the screening criteria are actually implemented in practice, positive screens were all patients who actually received an ECG within 10 min of arrival, regardless of what the protocol suggested.

#### 2.2.2. Screen 2: Per Clinical Protocol

Using the existing ED screening protocols, positive screens included patients aged 65 years or older or those reporting chest pain or another ACS-associated symptom upon arrival. We applied these criteria to identify those who would require an early ECG if the protocols were strictly followed.

#### 2.2.3. Screen 3: Predictive Model

Screening via the predictive model used the same ED arrival screening criteria as predictive independent variables, with a final diagnosis of ACS as the dependent variable. Additional modeling details are in the Statistical Analysis section below.

#### 2.2.4. Screen 4: Model-Augmented Human Practice

Across industries, there are recommendations for artificial intelligence and predictive modeling intended to augment human performance; these recommendations encourage the use of models to bridge gaps in human prediction activity [11,21]. We explored this approach by combining the positive screens from observed practice with those who were screened positive by the predictive model. In other words, patients who were positive on either observed practice (Screen 1) or screening by the model (Screen 3) were considered positive under model-augmented human performance (Screen 4). Hence, the model served as a fail-safe: it was able to add positive screens but not remove a positive screening determination from practice.

### 2.3. Demographic Subgroups

We included demographic subgroups for which the literature has previously described variation in the timeliness of care [4,8,9,10,11,22,23]. These include sex (male or female), age (18–29, 30–49, 50–64, 65–80, and >80), race (Asian, Black/African American, Native American, Alaskan, or Hawaiian/Pacific Islander, White, other race, and unknown/refused), ethnicity (Hispanic or Non-Hispanic), and language (English, Spanish, or other language).

### 2.4. Outcomes

For patients, the primary outcome was a final hospital diagnosis of ACS as per methods previously published and validated, which use international classification of disease billing codes [22,24,25]. We also examined cases of STEMI to understand the impact of false negatives and effective screening capture of ACS patients for an early ECG to identify those with STEMI.

For the screening methods, our outcomes are test characteristics: sensitivity (primary), specificity, number of ECGs required, number of missed ACS cases, and number of missed STEMI cases.

### 2.5. Statistical Analysis

We present counts and percent distributions for each demographic characteristic among the full study population, the subset with confirmed ACS, and the subset with confirmed STEMI. To quantify incidence, we also present the observed number of ACS and STEMI cases per 10,000 for each demographic group.

For the predictive model, we fit a logistic regression model that uses the same data elements available at registration (chest pain, other ACS symptoms, age, and sex) to predict ACS. We calculated each of the measurement outcomes directly from standard 2 × 2 contingency tables that compared screening status with true ACS status within each demographic subgroup. Appendix A contains more details on the modeling and calculation of measurement outcomes.

Comparing each test characteristic across all demographic groups would run a high risk of false discovery due to multiple comparisons. As a result, we did not include hypothesis testing in this descriptive exploration. Rather, we calculated each measure and provided 95% confidence intervals around sensitivity and specificity to facilitate interpretation.

## 3. Results

The cohort included 279,132 ED patients, of whom 1397 had a final hospital diagnosis of ACS. Of those with ACS, 225 had a final diagnosis of STEMI (Table 1). Compared to the total population of patients, those with ACS and STEMI were older, more often male, more often White or Asian, and less often Hispanic/Latino (columns D and G in Table 1). The highest rates for ACS and STEMI per 10,000 patients were observed among patients over 65, men, and non-Hispanic/Latino patients (Table 1).

### 3.1. Screening Criteria/Predictor Prevalence

Figure 1 includes the screening characteristics or predictors used to identify those at risk of ACS. The distribution of these characteristics across subgroups of age, sex, race, ethnicity, and language suggests variation in the distribution of risk across the subgroups. Chest pain (Panel A) showed the least variation across the age subgroups. It ranged from 8–10% in most subgroups, although it was markedly lower among the very young (5%, 18–29 years) and the elderly (6%, >80 years). Other ACS-associated symptoms (Panel B) were more prevalent than chest pain in the general population, ranging from 33% to 60% among the subgroups. They were more commonly present among those over the age of 65 (51–60%) and those who spoke neither English nor Spanish (50%).

The subgroups varied most in proportion over age 65 (Panel C), especially by race, language, and ethnicity. Among all ED patients, the White and Asian subgroups were the oldest, with one-third or more over age 65. The other racial groups were much younger, with only 16% to 18% over age 65. These sharp differences in age structure by race also appeared within the ACS and STEMI populations. Figure 2 shows that strictly applying the cutoff at age 65 would miss most cases of ACS (Panel A) and STEMI (Panel B) among Black, NAAH/PI, and other race patients. The concentration of cases in the under-65 age group is even more pronounced for STEMI than for ACS for all groups except White patients (Figure 2, with point estimates and confidence interval detail in Appendix A).

### 3.2. Screening Performance Outcomes

Table 2 shows the overall measurement properties for each screening method. Observed human practice (Screen 1) had 73% sensitivity and 78% specificity for ACS, with 22% of all patients requiring an ECG. This approach missed 27% of ACS cases and 15% of STEMI cases. In comparison, screening per clinical protocol (Screen 2) had the worst overall performance, with low sensitivity (56%) and correspondingly high proportions of missed ACS cases (45%) and missed STEMI cases (53%). The predictive model (Screen 3) had higher sensitivity (82%) than observed human practice, with similar specificity (78%). The model’s operating point was selected to require the same proportion of ECGs as human practice (22%), but it missed fewer ACS cases (18% as opposed to 27%); the proportion of STEMI cases missed was the same (15%). Finally, model-augmented human practice (Screen 4) had the highest sensitivity, at 92%, and the fewest missed cases of ACS (8%) and STEMI (4%). This screen would require 33% of patients to receive an ECG, however, as opposed to 22% under current practice. This would represent a 57% relative increase in the number needed to screen.

### 3.3. Screening Performance Variation across Subgroups

Figure 3 shows the sensitivity and specificity for all four screens among the total sample and by demographic subgroup. Screening by observed human practice (Screen 1) shows low variability in sensitivity across subgroups, ranging from 66% to 77%. This screen is slightly less sensitive in women compared to men, in Hispanic/Latino patients compared to others, among speakers of languages other than English, and in those over 80 years old compared to younger patients. For specificity, screening in observed human practice was lower among non-Hispanic patients, speakers of languages other than English or Spanish, and patients older than age 65.

The per clinical protocol screen had the lowest sensitivity in nearly all groups by far, though it had correspondingly higher specificity compared to the other screening methods. This screen had the most variability in sensitivity across the subgroups, driven by its sharp age cut-off at 65, such that sensitivity is 0% for those under 65, 92% for those aged 65 to 80, and 96% for those over 80. Sensitivity varied across racial groups from as low as 34% among NAAH/PI patients and 37% among Blacks to over 60% among White and Asian patients, driven largely by the different age structures by race as shown in Figure 2.

Screening by predictive model (Screen 3) exhibited higher sensitivity compared to observed practice for all subgroups except people under age 65, where its sensitivity was lower but comparable. Overall, the specificity of the predictive model follows a similar pattern by subgroup to observed practice by race, ethnicity, and language. It has lower specificity for men (71%) compared to women (84%), however, and its specificity at older ages is quite low (9% over age 80).

Finally, screening with model-augmented human performance (Screen 4) had the highest sensitivity across all subgroups and screening methods, except for the over-80 subgroup, where it matched the 99% sensitivity of the predictive model. This screen showed little variation by sex, race, or language and most variation by age, ranging from 87% for patients under 50 to 99% for the oldest group. Across most subgroups, however, this consistently high sensitivity is reflected in an approximately 10 percent reduction in specificity (compared to observed human practice) among most subgroups. There were marked reductions in specificity among those over 80 and those who did not speak English or Spanish. Further exploration revealed that this latter population was far older than the average (results not shown) and mostly consisted of Asian language speakers.

### 3.4. Screening Gain When Human Performance Is Augmented with the Model

Figure 4 reveals the measurement gains from combining observed human practice (Screen 1) with the predictive model (Screen 3). The resulting model-augmented human performance method (Screen 4) demonstrated superior sensitivity in our tests. Panel A shows the overall incidence of ACS by subgroup, which varies most strongly by age. As discussed above, the variations by race and language are partly driven by differences in the age structure of these groups. Panel B shows the percentage of patients who screened positive under Screen 4, which dictates the number of ECGs required. The predictive model added most screen positives to the older age categories, making up for the lower sensitivity of observed practice at those ages. In the over-80 age group, observed human practice identified 35% of patients as being at ACS risk, and the predictive model added a full 57% of patients to that, raising the screen positives to 92% total in that group under Screen 4. The rise in sensitivity from adding the predictive model to observed practice for ACS is shown in Panel C and for STEMI in Panel D. Setting aside the very young ages of 18–29, which had only 6 cases of ACS and only 1 STEMI, the predictive model was able to add to the sensitivity of all subgroups, ranging from +9% to +33% for ACS and +2% to +22% for STEMI.

## 4. Discussion

Despite using the same data to determine who should receive an early ECG, the four screening approaches yielded distinct results. Observed human practice (Screen 1) had reasonable sensitivity and was fairly equitable across demographic subgroups (Figure 3). In contrast, screening per clinical protocol (Screen 2) had marked inequity in performance, largely driven by subgroup differences in age structure (Figure 2). As anticipated, the predictive model (Screen 3) had higher sensitivity than observed practice (Screen 1) for many subgroups, and it was far higher among the oldest patients. Overall, model-augmented human performance (Screen 4) had the highest sensitivity of all the screens in every subgroup. While it increased the total number of ECGs from 22% to 33% of all ED patients, it managed to identify 92% of ACS cases and 96% of STEMI cases. Compared to observed human practice, this augmented model required ECGs on an additional 12% of patients, leading to an additional 19% of the ACS cases and 10% of the STEMI cases being captured.

### 4.1. Variation in Age Distributions by Subgroup

Many of our results showing inequity across the subgroups were driven by differences in age structure, particularly by racial group. Among the subgroups we considered, rates of ACS per 10,000 vary most strongly by age, ranging up to 120 for patients over 80—by far the highest incidence of any group (Figure 4). This strong relationship means that if a subgroup has a different age structure than another, as shown in Figure 2 for racial groups, then any screen based on age will behave differently in that group. For example, 33% of non-Hispanic/Latino patients in our sample were aged 65 or older, compared to only 13% of all Hispanic/Latino patients. In an unexpected skew, a full 64% of “other” language speakers were 65 or older, compared to just 26% of English speakers and 20% of Spanish speakers. The variations in age structures by group are the product of social, medical, and historical circumstances that are well beyond the scope of this study. The consequence of potentially missed ACS screening is de facto inequities that arise in practice due to the race, ethnicity, and language of patients. Fortunately, multivariable models that include these types of characteristics, including nonlinear terms such as interactions, may result in better sensitivity and specificity of screening methods. The fairness and consequences of including race in clinical prediction models are widely debated [26,27]; in future research, we plan to investigate this question for our specific use case.

### 4.2. Challenge of Screening via Current Human Practice

Although observed practice did not have the highest sensitivity, it exhibited limited variability across demographic groups and fewer missed STEMI cases compared to the clinical protocols alone. We surmise that those performing manual screening leverage or respond to other available information, such as visual or linguistic cues about a patient’s condition, that influence their selection of patients [28].

### 4.3. Advantages of Predictive Modeling

The predictive model for Screen 3 included the same data elements used in observed care and in the clinical protocol (age, chest pain, and other ACS symptoms), yet it resulted in higher sensitivity in most subgroups. Its superior sensitivity to Screen 2 was largely due to the treatment of age: model 3 encoded age as a continuous variable rather than a blunt dichotomy at age 65. We cannot compare the working of the model to observed care because we have no insight into how the registration clerks made their decisions. We can, however, show that the model and the clerks identified somewhat different groups of patients for early ECGs.

### 4.4. Added Value of Model-Augmented Human Practice

Model-augmented human practice (Screen 4) had the highest sensitivity across all groups and above all other approaches. This was accompanied by a reduction in specificity across all groups due to a notable increase in false positives. Although increased testing from false positive screens is not desired, missed screening (false negative) delays diagnosis and is far less acceptable given the medical gravity of the delay in care. Indeed, prior work has suggested over-testing may be appropriate in certain populations that carry high risk due to the associated co-morbidity burden and communication delays associated with the need for translation [29]. Furthermore, additional ECGs are a relatively low-cost, rapidly performed diagnostic test.

We found that adding the predictive model to clinical practice would trigger ECGs for an additional 11% of patients (Table 2) while increasing sensitivity by 19 and 10 percentage points for ACS and STEMI, respectively (Figure 4, Panels C and D). This would result in a 72% [(374 − 106)/374] increase in the detection of ACS cases, and a 70% [(33 − 10)/33] increase in the detection of STEMI cases. (Table 2).

Augmented practice counterbalanced the demographic groups for which observed practice had lower sensitivity and greater variation in sensitivity. In particular, the patients added by Screen 3 were predominantly women, over the age of 80, and those who did not speak English or Spanish. This suggests that collaboration between humans and modeling can deliver better screening performance as well as improved equity. This accords with prior studies that observed that humans and predictive modeling have distinct strengths. Whereas models can be more consistent, humans can be more intuitive, emotional, and culturally sensitive in ways that we are just learning to quantify [16]. This suggests that the synergy of strengths from combining human performance with predictive modeling can optimize benefits for care delivery.

## 5. Limitations

The importance of considering chest pain as a predictive chief complaint and the challenge of balancing the consideration of ACS-associated chief complaints with a high false negative rate are universal. Thus, the patterns we observe are likely to be generalizable. Nevertheless, there are limitations that should be noted when interpreting our results.

Our study used a large sample of patients from a diverse patient population to conceptually explore ACS prediction performance, variation, and equity, but our subjects were drawn from a single-center study. Since the distribution of patient demographics of our study population may vary from other EDs, the prevalence and distribution of the predictive characteristics therefore may vary at other sites.

Our observations of Native American, Alaskan, or Hawaiian/Pacific Islander patients are novel and likely permitted by this being a population with notable representation in our regions; as such, we note that analogous disparities seen among black patients and women have been previously reported [4].

Finally, we report only one way in which models can augment care: by having a model run concurrently and augment human decision-making. There are other ways for models to augment human performance. For example, models being interactive for the screener, such as models suggesting specific questions that the person screening can ask to better ascertain the patient’s condition. These additional explorations could be the basis of future work.

We present this study as a conceptual simulation of screening options including augmented human practice using retrospective data. The next steps to better understand the implications of these models would be to perform prospective research through the implementation of these models into electronic health records. The move from desktop to bedside requires extensive work but will be critical to the future success of predictive models in healthcare. To build on our findings, we encourage research into these questions using prospective designs to account for data that may be missing at the time of risk calculation that are completed at a later time during the encounter. Our data were collected well after the 10-min time window to perform an ECG. The data could have been entered after those 10 min and therefore not appear as missing in our dataset. Therefore, it will be important to prospectively capture this data to identify whether the appropriate data are available within 10 min of arrival for the predictive model to be usable. This kind of time-sensitive missingness is not well represented in archived clinical data, but patterns associated with systematically missing data vs. ‘missingness at random’ may influence the performance of structured screening approaches more than human-driven practice [30].

## 6. Conclusions

We found that augmented screening (observed practice supplemented with the predictive model) had the best sensitivity (92.4%) and missed fewer ACS patients (7.6% vs. 26.8%) and STEMIs (4.4% vs. 14.7%) than observed human practice. It also showed little variation in sex, race, ethnicity, language, and age, demonstrating improved equity compared to other approaches. Augmented screening increased sensitivity by 20.8%, required ECGs to be performed on an additional 11.1% of patients, and reduced specificity by 14%. The increase in ECGs is likely within the tolerable range, however, given the marked gain in ACS case detection. Furthermore, it suggests that combining human and model screening––rather than relying on either one alone––may improve both overall performance and equity in ACS screening.

## Figures and Tables

**Figure 1 diagnostics-13-02053-f001:**
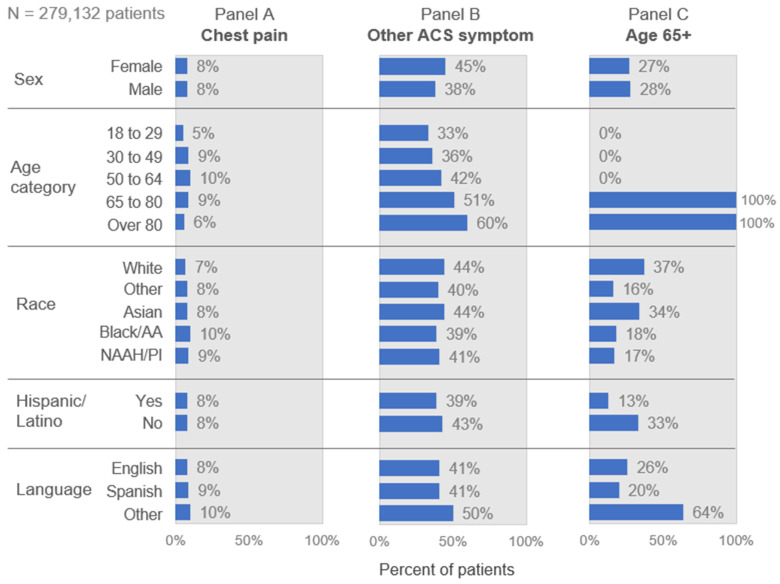
Prevalence of key screening criteria/predictors by demographic group. NAAH/PI = Native American, Alaskan, or Hawaiian/Pacific Islander. This figure presents the prevalence of the screening criteria used as guidance for human practice and as predictors in the model.

**Figure 2 diagnostics-13-02053-f002:**
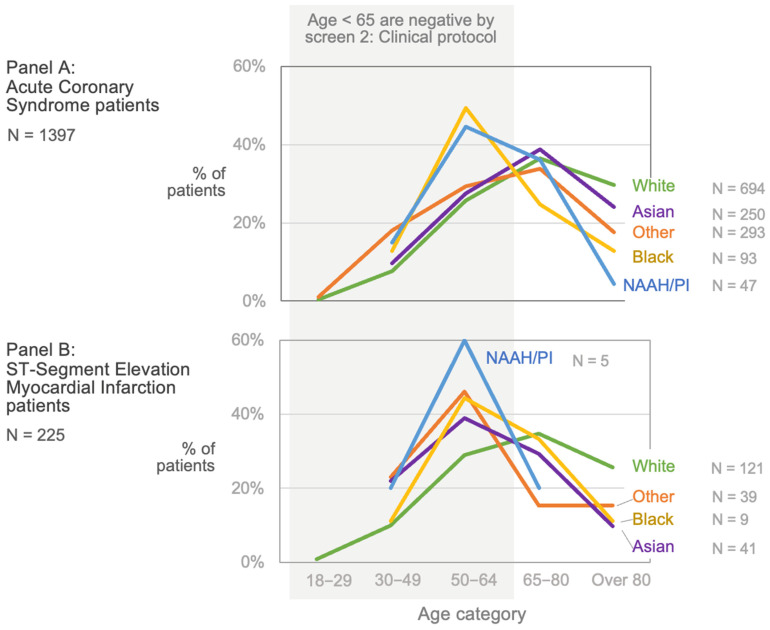
Acute coronary syndrome (ACS) and STEMI patient age by race. Using ‘age >65 years’ as a screening criterion to identify those at risk of ACS introduces substantial inequity. Panel **A** shows that Black and NAAH/PI ACS patients are generally younger than White, Asian, or other race patients, and many are under age 65. The concentration of cases in the younger age group is even more pronounced among STEMI patients (Panel **B**), including those of Asian or other race. The sharp cut-off at age 65 for screening inadvertently disadvantages sub-groups with younger age structures, which can fall along racial lines in the US. NAAH/PI = Native American, Alaskan, or Hawaiian/Pacific Islander.

**Figure 3 diagnostics-13-02053-f003:**
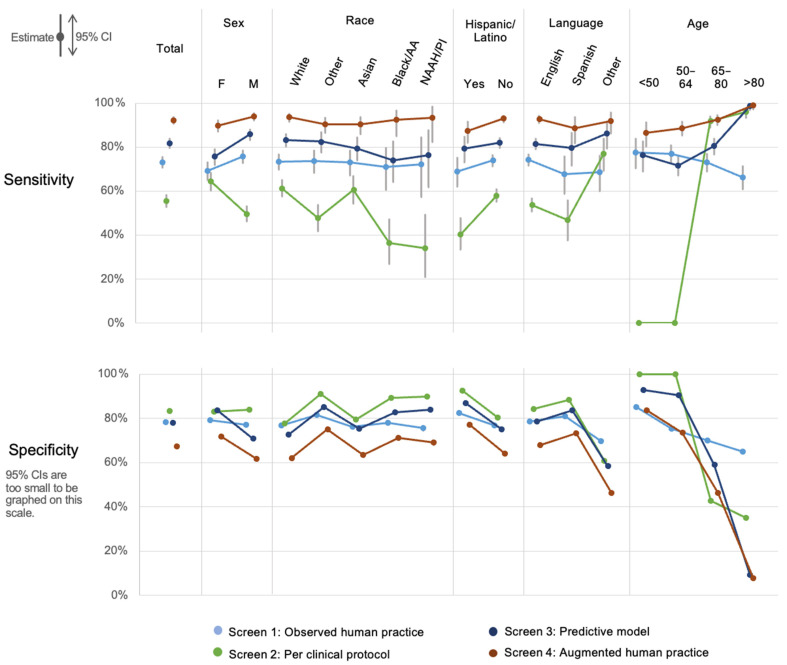
Sensitivity and specificity of screening approaches for acute coronary syndrome by demographic subgroup. NAAH/PI = Native American, Alaskan, or Hawaiian/Pacific Islander. screening per clinical protocol had the widest variation in sensitivity across the demographic groups, and observed human practice had lower variation. When observed practice was augmented by the predictive model, sensitivity increased for all sub-groups except those >80 years of age, which already had a sensitivity of nearly 100%. Increased ACS case detection with the model-augmented human practice screen results in a lower specificity by about 10 percentage points compared to observed human practice alone. Sensitivity and specificity are not equally important, however, given the high intolerance for missed ACS and the low burden of performing additional early ECGs.

**Figure 4 diagnostics-13-02053-f004:**
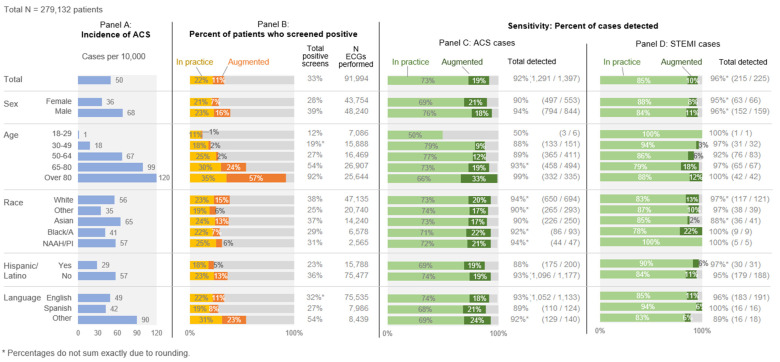
ACS and STEMI incidence, positive ACS screens, and true positive case capture. NAAH/PI = Native American, Alaskan, or Hawaiian/Pacific Islander. Panel A presents the incidence of ACS across the demographic subgroups, illustrating differential risk. Panel B presents the percent of the total ED population who screened positive under observed human practice (yellow) and the positive screens that would be added under the model-augmented human practice approach (orange). Panels C and D present the proportion of ACS patients in each subgroup identified via observed human practice (green) and those that would be added with model-augmented human practice screening (dark green).

**Table 1 diagnostics-13-02053-t001:** Demographic characteristics of the total emergency department compared to ACS and STEMI patients with prevalence.

		Total		ACS Patients			STEMI Patients		
	Column:	A	B	C	D	E	F	G	H
		N	Column %	N	Column %	Cases per 10K	N	Column %	Cases per 10K
Total		279,132	100%	1397	100%	50	225	100%	8.1
Age Group	18 to 29	58,407	21%	6	0%	1	1	0%	0.2
	30 to 49	81,811	29%	151	11%	18.5	32	14%	3.9
	50 to 64	61,228	22%	411	29%	67.1	83	37%	13.6
	65 to 80	49,879	18%	494	35%	99	67	30%	13.4
	Over 80	27,807	10%	335	24%	120.5	42	19%	15.1
Sex	Female	154,485	55%	553	40%	35.8	66	29%	4.3
	Male	124,647	45%	844	60%	67.7	159	71%	12.8
Race (In order of group size)	White	123,806	44%	694	50%	56.1	121	54%	9.8
	Other	82,869	30%	293	21%	35.4	39	17%	4.7
	Asian	38,631	14%	250	18%	64.7	41	18%	10.6
	Black	22,596	8%	93	7%	41.2	9	4%	4
	NAAH/PI	8216	3%	47	3%	57.2	5	2%	6.1
	Unknown/ Refused	2745	1%	18	1%	65.6	8	4%	29.1
Hispanic/ Latino	Yes	68,534	25%	200	15%	29.2	31	14%	4.5
	No	207,776	75%	1177	85%	56.6	188	86%	9
Language	English	233,556	84%	1133	81%	48.5	191	85%	8.2
	Spanish	29,704	11%	124	9%	41.7	16	7%	5.4
	Other	15,631	6%	140	10%	89.6	18	8%	11.5

NAAH/PI = Native American, Alaskan, or Hawaiian/Pacific Islander.

**Table 2 diagnostics-13-02053-t002:** Comparing screening approaches to identify patients at risk of ACS for an early ECG: sensitivity, specificity, ECGs required, and missed ACS and STEMI screens.

	Among All Patients (N = 279,132)				Among ACS Cases (N = 1397)		Among STEMI Cases (N = 225)	
Screen	Sensitivity (95% CI)	Specificity (95% CI)	Positive Screens					
			N Requiring ECGs	% Requiring ECGs	N Missed	% Missed	N Missed	% Missed
1. Observed practice	73.2% (70.8–75.5%)	78.3% (78.2–78.5%)	61,156	21.9%	374	26.8%	33	14.7%
2. Screening by protocol	55.5% (52.9–58.2%)	83.4% (83.3–83.6%)	46,818	16.8%	621	44.5%	119	52.9%
3. Screening by the predictive model	81.9% (79.8–83.9%)	78.1% (77.9–78.2%)	62,011	22.2%	253	18.1%	33	14.7%
4. Observed practice augmented with a predictive model	92.4% (90.9–93.7%)	67.3% (67.2–67.5%)	91,994	33.0%	106	7.6%	10	4.4%

Screening per protocol had the lowest sensitivity at 55.5%. Observed practice augmented by the predictive model achieved the highest sensitivity of 92.4% and missed the fewest ACS and STEMI cases.

## Data Availability

A de-identified version of this data may be made available upon request from the corresponding author, for the conduct of research, after review of the requestor’s intent and adherence with data sharing regulations.

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
