# Peer review of "Maximizing Equity in Acute Coronary Syndrome Screening across Sociodemographic Characteristics of Patients"

_diagnostics, 2023, doi:10.3390/diagnostics13122053_

Round 1
Reviewer 1 Report
The submitted manuscript presents data from a study aiming to maximize equity of Acute Coronary Syndrome (ACS) screening by screening patients with suspected ACS for an early electrocardiogram (ECG). The latter part is questionable. Please let me explain why.
ECG is probably the simplest, most affordable, cheapest diagnostic method in cardiology. It is surprising why screening with 4 approaches is needed to decide to take an ECG. I am certainly not against artificial intelligence (AI). But applying it in this situation seems to me excessive and unreasonable. It is certain and indisputable that every person presenting to the ED with chest pain, regardless of gender, age or race, should have an ECG because the procedure takes so little time and cost.
The conclusion of the article about the advantages of combining AI with the human factor is predictable and correct. The study was done flawlessly, so my doubts may be unfounded. However, another reader may also find it excessive to use AI to solve the issue of ECG registration. In this regard, it may be necessary to add some reasoning on this matter.
In general, the presented work has been done clearly and provides appropriate results. It is relevant for the field, scientifically sound, and presented in a well-structured manner. The performed statistical analysis is convincing. I have no significant remarks. As a result, the manuscript may be recommended for publication.
Author Response
REVIEWER 1
The submitted manuscript presents data from a study aiming to maximize equity of Acute Coronary Syndrome (ACS) screening by screening patients with suspected ACS for an early electrocardiogram (ECG). The latter part is questionable. Please let me explain why.
Reviewer 1 Comment 1: ECG is probably the simplest, most affordable, cheapest diagnostic method in cardiology. It is surprising why screening with 4 approaches is needed to decide to take an ECG. I am certainly not against artificial intelligence (AI). But applying it in this situation seems to me excessive and unreasonable. It is certain and indisputable that every person presenting to the ED with chest pain, regardless of gender, age or race, should have an ECG because the procedure takes so little time and cost.
Response: The reviewer is correct in assessing the low cost of ECGs, and that they are done quite cheaply. However, performing diagnostic testing before queueing for physician evaluation in an emergency department comes at the cost of slowing down the entry to care for other patients. In addition, each ECG performed must be interpreted immediately by a physician who is caring for other patients who likely have more common emergency conditions. If the interpretation is concerning, the patient is immediately seen by a physician for care before others who may be waiting. Thus, the selection of arriving patients for potential ACS, before physician evaluation, is a significant clinical care delivery challenge. Furthermore, our model takes into account that every patient presenting with chest pain receives an ECG. However, it is the more subtle cases where patients present with symptoms other than chest pain, but are still ACS equivalents, that we are attempting to capture. We have added additional clinical context to the “Introduction” to make this challenge more clear.
Reviewer 1 Comment 2: The conclusion of the article about the advantages of combining AI with the human factor is predictable and correct. The study was done flawlessly, so my doubts may be unfounded. However, another reader may also find it excessive to use AI to solve the issue of ECG registration. In this regard, it may be necessary to add some reasoning on this matter.
Response: Duly noted. In response we have added the detail we described in the response above to the “Introduction.”
Reviewer 1 Comment 3: In general, the presented work has been done clearly and provides appropriate results. It is relevant for the field, scientifically sound, and presented in a well-structured manner. The performed statistical analysis is convincing. I have no significant remarks. As a result, the manuscript may be recommended for publication.
Response: Thank you for consideration and your questions encouraged us to present the rationale for our investigation.
Reviewer 2 Report
The authors retrospectively examined the diagnostic accuracy of four different screening methods for diagnosing ACS and STEMI. While the study involved a large number of cases and is interesting, there are several concerns.
Major comments:
This study focuses on 1,397 cases diagnosed with ACS out of 279,134 cases who visited the Emergency Department. There is a possibility that ACS cases were missed among the remaining 277,737 cases, and therefore the accuracy of the study itself is not guaranteed.
P3,L119; Although ACS was diagnosed based on the international classification of disease billing code, and the researchers did not clinically confirm the diagnosis. Ideally, the diagnosis should be confirmed clinically according to the fourth universal definition of myocardial infarction (Circulation 2018;138:e618-e51).
Minor comments:
P1,L12; In the first mention, ED should not be an abbreviation.
P2,L78; The name of the hospital and the country should be specified.
P2,L78; The details of the Institutional Review Board's name and approval number should be provided.
Author Response
REVIEWER 2
The authors retrospectively examined the diagnostic accuracy of four different screening methods for diagnosing ACS and STEMI. While the study involved a large number of cases and is interesting, there are several concerns.
Major Comments
Reviewer 2 Comment 1: This study focuses on 1,397 cases diagnosed with ACS out of 279,134 cases who visited the Emergency Department. There is a possibility that ACS cases were missed among the remaining 277,737 cases, and therefore the accuracy of the study itself is not guaranteed.
Response: As will all research involving the secondary use of clinical data, studies must be rigorously performed to inform practice, and can include very large numbers of patients, but will fall short of perfection. However, the methodology in establishing our gold standard was to use ICD-9 and 10 codes for the diagnosis of STEMI, NSTEMI, and unstable angina, were validated and used in prior work listed below:
- Bunney G, Sundaram V, Graber-Naidich A, Miller K, Brown I, McCoy AB, Freeze B, Berger D, Wright A, Yiadom MY. Beyond chest pain: Incremental value of other variables to identify patients for an early ECG. The American Journal of Emergency Medicine. 2023 May 1;67:70-8.
- Wu J, Fan Y, Zhao W, Li B, Pan N, Lou Z, Zhang M. In‐Hospital Outcomes of Acute Myocardial Infarction With Essential Thrombocythemia and Polycythemia Vera: Insights From the National Inpatient Sample. Journal of the American Heart Association. 2022 Dec 20;11(24):e027352.
- Yiadom MY, Gong W, Patterson BW, Baugh CW, Mills AM, Gavin N, Podolsky SR, Salazar G, Mumma BE, Tanski M, Hadley K. Fallacy of Median Door‐to‐ECG Time: Hidden Opportunities for STEMI Screening Improvement. Journal of the American Heart Association. 2022 May 3;11(9):e024067.
- Yiadom MY, Olubowale OO, Jenkins CA, Miller KF, West JL, Vogus TJ, Lehmann CU, Antonello VD, Bernard GR, Storrow AB, Lindsell CJ. Understanding timely STEMI treatment performance: a 3‐year retrospective cohort study using diagnosis‐to‐balloon‐time and care subintervals. Journal of the American College of Emergency Physicians Open. 2021 Feb;2(1):e12379.
- Yiadom MY, Baugh CW, McWade CM, Liu X, Song KJ, Patterson BW, Jenkins CA, Tanski M, Mills AM, Salazar G, Wang TJ. Performance of emergency department screening criteria for an early ECG to identify ST‐segment elevation myocardial infarction. Journal of the American Heart Association. 2017 Feb 23;6(3):e003528.
- Yiadom MY, Mumma BE, Baugh CW, Patterson BW, Mills AM, Salazar G, Tanski M, Jenkins CA, Vogus TJ, Miller KF, Jackson BE. Measuring outcome differences associated with STEMI screening and diagnostic performance: a multicentred retrospective cohort study protocol. BMJ open. 2018 May 1;8(5):e022453.
- Bloos SM, Kaur K, Lang K, Gavin N, Mills AM, Baugh CW, Patterson BW, Podolsky SR, Salazar G, Mumma BE, Tanski M. Comparing the Timeliness of Treatment in Younger vs. Older Patients with ST-Segment Elevation Myocardial Infarction: A Multi-Center Cohort Study. The Journal of Emergency Medicine. 2021 Jun 1;60(6):716-28.
We continue to strengthen our methods as we learn more about clinical diagnosis data. We have confidence that we have captured ACS in a rigorous manner that permits us to perform the comparisons in our analyses with validity and reliability.
Reviewer 2 Comment 2: P3,L119; Although ACS was diagnosed based on the international classification of disease billing code, and the researchers did not clinically confirm the diagnosis. Ideally, the diagnosis should be confirmed clinically according to the fourth universal definition of myocardial infarction (Circulation 2018;138:e618-e51).
Response: This is most definitely the case with a clinical chart review or prospective cohort study where individual patients are selected for inclusion. Being that this was an epidemiological analysis using large volume electronic health record data, we did not employ chart review for all 279K patients in the analysis or 1397 identified ACS cases. However, we did employ several methodological steps to have confidence in our diagnostic reference standard:
- We assessed whether there were ACS cases among those discharged, and did not find any which is consistent with recommended practice.
- We spot reviewed the charts of all ED visits with an ACS diagnosis, but no evidence of troponin or ECG testing that would be used to make the diagnosis. We found that all had documented evidence of testing prior to arrival to support the diagnosis.
- The ICD codes we used to define ACS (include those for STEMI, NSTEMI and UA have been validated and used in other studies as is listed in our response to Reviewer 2’s Comment 1)
However, we are continuing to build upon our work in the project. In our next project related to this work, we are exploring methods to take a deeper look within patient data and the diagnostic information to see how this may impact our models going forward.
Minor Comments:
Reviewer 2 Comment 3: P1,L12; In the first mention, ED should not be an abbreviation.
Response: Corrected
Reviewer 2 Comment 4: P2,L78; The name of the hospital and the country should be specified.
Response: Corrected
Reviewer 2 Comment 5: P2,L78; The details of the Institutional Review Board's name and approval number should be provided.
Response: Corrected
Reviewer 3 Report
Bunney et al. conducted a study aimed at triage of ACS patients. The biggest limitation of the study is its retrospective design, but in consideration of large sample the reader could benefit from reading this study. The methodology is overall acceptable, although it could be presented somewhat more clear.
Why previous ACS or presence of any previous "heart condition" was not considered in modeling, that is a readily available information that would undoubtedly improve sensitivity and specificity of the method.
A major issue in this paper is discussion, as discussion is written like a very large conclusion. The fact is that the authors clarified their results a bit in this section, but what it lacks is critical comparison with the available data and future implications of this method (How to improve it? How to implement it? What are the challenges?).
Minor points:
Background subheading is not needed
Line 112. Something is wrong with this sentence
Minor edits are needed
Author Response
REVIEWER 3
Bunney et al. conducted a study aimed at triage of ACS patients. The biggest limitation of the study is its retrospective design, but in consideration of large sample the reader could benefit from reading this study. The methodology is overall acceptable, although it could be presented somewhat more clear.
Reviewer 3 Comment 1: Why previous ACS or presence of any previous "heart condition" was not considered in modeling, that is a readily available information that would undoubtedly improve sensitivity and specificity of the method.
Response: Several factors such as a history of prior ACS, coronary artery disease, cardiovascular disease condition, prior coronary artery bypass graft surgery, Framingham risk factors such as hyperlipidemia or diabetes, and other comorbidities may be predictive of a new ACS episode. This information is readily available to the physicians caring for the patient, and perhaps even the triaging nurse upon review of the medical record, but it is not available upon arrival. Obtaining it requires a discussion with the patient, or review of their medical record. The point at which this screening is performed is when a chart is created, often by a non-clinical staff member, to support the emergency visit upon arrival, and before a clinical provider has encountered the patient. Waiting for the clinical provider will extend a decision to obtain an early ECG beyond the goal of acquiring the ECG within 10 minutes. We have added additional details in response to Reviewer 1’s Comment 2, to the 2nd , 3rd, and 4th paragraphs of the Introduction to better present the clinical context of emergency department ACS screening.
There is concurrent work, both within our group and others, to making prior information from a patient’s historical medical record instantly available upon arrival. We found that 50-75% of patients arriving to many emergency departments have a record with the associated health system. As a result, prior information from their record could be useful. A 2-tiered screening model that includes information collected upon arrival, and previously existing co-morbidity (or other information), would augment assessment of risk for a majority of patients. However, there are technical limitations within the majority of electronic health record systems to enable this. In this paper we establish the value and limitations of predictive modeling augmenting human decisions. It is our hope that this will facilitate a pathway to addressing the limitations.
There is additional work needed to address the bias that may be introduced by a second tier of prediction. When there is missing data in 25-75% of the patients, their predictions are likely to be less precise and may increase the likelihood of a disparity. This equity-oriented work is an area for future work.
Reviewer 3 Comment 2: A major issue in this paper is discussion, as discussion is written like a very large conclusion. The fact is that the authors clarified their results a bit in this section, but what it lacks is critical comparison with the available data and future implications of this method (How to improve it? How to implement it? What are the challenges?).
Response: We have clarified our thoughts on the future implications of this method in the Discussion section.
Minor points:
Reviewer 3 Comment 3: Background subheading is not needed
Response: Removed
Reviewer 3 Comment 4: Line 112. Something is wrong with this sentence
Response: Edited for clarity.
Reviewer 4 Report
Thank you for asking me to review this manuscript This is undoubtedly an interesting paper, each cardiologist should read. The manuscript is well written and data are well described, although the known study limitations.
Considering the main topic of this paper, the conclusions are consistent with the evidences reported. The references are appropriate. Combining human and model screening surely improves the performance and equity in ACS screening and future research could be useful to guide combined diagnostic approach.
Author Response
REVIEWER 4
Thank you for asking me to review this manuscript This is undoubtedly an interesting paper, each cardiologist should read. The manuscript is well written and data are well described, although the known study limitations.
Reviewer 4 Comment 1: Considering the main topic of this paper, the conclusions are consistent with the evidences reported. The references are appropriate. Combining human and model screening surely improves the performance and equity in ACS screening and future research could be useful to guide combined diagnostic approach
Response: Thank for this very favorable review and assessment. We conducted this investigation with rigor to be most informative for emergency cardiovascular care.
-------------------------------------------------------------------------------------
Round 2
Reviewer 2 Report
My suggestions have all been appropriately corrected.